# Low Inflammatory Stimulus Increases D2 Activity and Modulates Thyroid Hormone Metabolism during Myogenesis In Vitro

**DOI:** 10.3390/metabo12050416

**Published:** 2022-05-06

**Authors:** Thamires Siqueira de Oliveira, Marilia Kimie Shimabukuro, Victoria Regina Siqueira Monteiro, Cherley Borba Vieira Andrade, Anita Boelen, Simone Magagnin Wajner, Ana Luiza Maia, Tania Maria Ortiga-Carvalho, Flavia Fonseca Bloise

**Affiliations:** 1Laboratory of Translational Endocrinology, Carlos Chagas Filho Institute of Biophysics, Universidade Federal do Rio de Janeiro, Rio de Janeiro 21941-902, Brazil; tsoliveira@biof.ufrj.br (T.S.d.O.); kimie@biof.ufrj.br (M.K.S.); victoriasiqmont@biof.ufrj.br (V.R.S.M.); cherley.andrade@uerj.br (C.B.V.A.); taniaort@biof.ufrj.br (T.M.O.-C.); 2Department of Histology and Embryology, Roberto Alcantara Gomes Institute of Biology, Universidade do Estado do Rio de Janeiro (UERJ), Rio de Janeiro 20551-030, Brazil; 3Endocrine Laboratory, Department of Clinical Chemistry, Amsterdam University Medical Center, Location AMC, 1105 AZ Amsterdam, The Netherlands; a.boelen@amsterdamumc.nl; 4Thyroid Unit, Endocrine Division, Hospital de Clínicas de Porto Alegre, Universidade Federal do Rio Grande do Sul, Porto Alegre 90035-903, Brazil; simone.wajner@ufrgs.br (S.M.W.); almaia@ufrgs.br (A.L.M.)

**Keywords:** myoblast, C2C12, deiodinase, triiodothyronine, myogenic differentiation, bacterial lipopolysaccharide, inflammation

## Abstract

Thyroid hormone (TH) signaling controls muscle progenitor cells differentiation. However, inflammation can alter muscle TH signaling by modulating the expression of TH transporters (*Slc16a2)*, receptors (*Thra1*), and deiodinase enzymes (*Dio2* and *Dio3*). Thus, a proinflammatory environment could affect myogenesis. The role of a low-grade inflammatory milieu in TH signaling during myogenesis needs further investigation. Herein, we aimed to study the impact of the bacterial lipopolysaccharide (LPS)-induced inflammatory stimulus on the TH signaling during myogenesis. C2C12 myoblasts differentiation was induced without (CTR) or with 10 ng/mL LPS presence. The myoblasts under LPS stimulus release the proinflammatory cytokines (IL-6 and IL-1β) and chemokines (CCL2 and CXCL-1). LPS decreases *Myod1* expression by 28% during the initial myogenesis, thus reducing the myogenic stimulus. At the same time, LPS reduced the expression of *Dio2* by 41% but doubled the D2 enzymatic activity. The late differentiation was not affected by inflammatory milieu, which only increased the *Slc16a2* gene expression by 38%. LPS altered the intracellular metabolism of TH and reduced the initial myogenic stimulus. However, it did not affect late differentiation. Increased intracellular TH activation may be the compensatory pathway involved in the recovery of myogenic differentiation under a low-grade inflammatory milieu.

## 1. Introduction

Myogenesis is a crucial process for skeletal muscle regeneration. The myogenic activity depends on the stimulus that leads to activation, proliferation, and differentiation of satellite cells into myoblasts [1]; these cells proliferate and differentiate into myocytes. Then, myocytes fuse to form multinucleated cells called myotubes which mature to form new myofibers [2]. Intrinsically, the hierarchical expression of a transcription factors family known as myogenic regulatory factors (MRFs) regulate myogenesis [3]. However, extrinsic factors such as thyroid hormones (TH) are also important signals that regulate myogenic differentiation [4]. Additionally, T3 controls the expression of MRF factors such as Myod and Myog [5,6,7].

Deiodinase type 2 (D2) and type 3 (D3) are the primary regulators of intracellular TH concentration in skeletal muscle [8]. An increase in D2 elevates intracellular T3 levels, while the D3 activity culminates in a decrease in T3 levels [8]. D2 and D3 expressions are finely controlled during myogenesis [9]. Low intracellular TH levels are necessary to survive and satellite cell proliferation [10]. Thus, proliferating satellite cells present high D3 expression [10]. After the satellite cell proliferative phase, D3 expression is reduced while D2 expression increases to promote cell differentiation [11,12]. The increase in D2 activity during the myoblasts and myocytes differentiation is associated with the MyoD and myogenin induction by T3 [11]. MyoD and myogenin regulate late myoblasts differentiation and induce myocytes to express structural proteins such as MyHC and differentiate into myotubes [11]. In addition to deiodinases, TH receptor TRα and TH transporters MCT8 and OATP1C1 proved to be essential for fine control of TH signaling as well as for complete muscle regeneration [13,14]. Therefore, disturbances in the expression of deiodinases, receptors, transporters, and consequently in TH signaling during myogenesis can compromise the regeneration and maintenance of muscle tissue. 

Inflammation can disturb the TH metabolism in both clinical and experimental settings [15]. Altered TH metabolism during illness is known as nonthyroidal disease syndrome (NTIS) or low T3 syndrome, which is characterized by concomitant low serum levels of TH and normal serum TSH concentration [15]. Moreover, profound changes in peripheral TH signaling in several target organs such as the liver and skeletal muscle are observed [15]. Besides the modulation of serum TH concentrations, TH signaling in skeletal muscle changes during NTIS [15].

In 2016, our group demonstrated that bacterial sepsis and chronic aseptic inflammation influenced the expression of D2, D3, and TRα in the murine diaphragm [16]. An increase in D3 and a reduction in D2 were observed in the diaphragm of septic mice, thereby reducing intracellular TH signaling [16]. On the other hand, chronic inflammation led to a reduction in D3 expression in the diaphragm. These changes in TH tissue metabolism were accompanied by changes in the expression of MyHC, which could be associated with diaphragm dysfunctions commonly seen in septic patients [16,17]. Subsequently, our group evaluated muscle TH metabolism in three NTIS models: acute inflammation, chronic inflammation, and the severe pneumonia sepsis model [16,17,18]. Comparing the experimental models, it was observed that muscle TH metabolism is differently affected by the disease′s type and severity [18].

Systemic inflammation induces loss of muscle mass. Septic patients and patients with several chronic diseases can develop myopathy [19]. The decrease in muscle function is associated with high proinflammatory cytokine serum levels [20]. Furthermore, a lipopolysaccharide (LPS)-induced proinflammatory milieu impaired C2C12 myoblast differentiation [21]. Although the inflammatory milieu is responsible for altered TH signaling in skeletal muscle, the implications of these changes in myogenesis have not been evaluated. Reduced myogenesis is a factor that can lead to myopathy in critically ill patients. This work aims to evaluate the impact of the inflammatory milieu on TH signaling during myogenic differentiation in vitro. Understanding the relationship between TH signaling and the inflammatory milieu can contribute to the recovery of myopathies and injuries caused by inflammatory diseases.

## 2. Results

### 2.1. C2C12 Myoblasts Produce Cytokines and Chemokines in Response to Bacterial Lipopolysaccharide

To confirm if the LPS dose used (10 ng/mL) was able to induce a proinflammatory milieu in vitro, we measured the proinflammatory cytokines (IL-6 and IL-1β) and chemokines concentration (CCL-2 and CXCL-1) in the C2C12 myoblast conditioned medium at 24 h after differentiation induction (72 h after the first challenge with LPS). We observed a significant increase in IL-6 (Figure 1A) and IL-1β (Figure 1B) concentration in the supernatant of LPS treated myoblasts. Chemokines CCL-2 (Figure 1C) and CXCL-1 (Figure 1D) concentrations also increased in the supernatant of C2C12 myoblasts treated with LPS (10 ng/mL).

### 2.2. LPS Treatment Decreases the Initial Myogenic Stimulus but Does Not Affect Late Differentiation

To assess whether the proinflammatory environment induced by LPS was able to affect myogenic differentiation, we evaluated the gene expression of different myogenic markers.

Initially, we evaluated the myogenic master transcription factor, *Myod*, gene expression during the early differentiation. We observed that *Myod1* expression was significantly reduced by LPS treatment (*p* = 0.009, Figure 2A), suggesting a reduction in the initial myogenic stimulus at 24 h.

To examine if the reduction in the initial myogenic stimulus induced by LPS impacted late differentiation (96 h), we evaluated the expression of *Myog* (myogenin), an MRF induced by *Myod1,* and highly expressed in myocytes [3]. LPS administration did not alter the *Myog* gene expression at late differentiation (*p* = 0.59, Figure 2B). Further, we evaluated the expression of the genes encoding MyHC I and IIb, *Myh7*, and *Myh4*, respectively. *Myh7* and *Myh4* mRNA expression were similar in control and LPS myocytes (*p* = 0.36, Figure 2C and *p* = 0.22, Figure 2D).

### 2.3. Inflammation Differently Affects TH Signaling during Initial and Late Myogenic Differentiation

Since T3 stimulates *Myod* and *Myog* expression, we next investigated if the proinflammatory environment would affect TH signaling in myoblasts during differentiation. We initially investigated TH signaling in the early differentiation stage (24 h post-differentiation induction). The TH transporter MCT8 (*Slc16a2)* expression was not altered by LPS treatment (*p* = 0.22, Figure 3A). Next, we evaluated *Dio2* expression to assess the intracellular TH metabolism. LPS treatment significantly reduced *Dio2* expression (*p* = 0.002, Figure 3B). However, D2 activity was increased by LPS treatment, suggesting an increase in intracellular T3 production (*p* = 0.0001, Figure 3C). Then, we evaluated the expression of the TH receptor predominantly expressed in muscle, *Thra1*, which showed a tendency to decrease in LPS group (*p* = 0.067, Figure 3D). To better understand the responsiveness to T3, we analyzed the expression of *Hr*, a T3 responsive gene. Despite the increase in D2 activity, *Hr* expression was not changed by LPS during the initial myogenic differentiation (*p* = 0.58, Figure 3E).

Subsequently, we evaluated TH signaling 96 h post differentiation induction during the late myocyte differentiation stage. LPS treatment increased *Slc16a2* mRNA levels (*p* = 0.04, Figure 4A), suggesting increased TH flow through the sarcolemma. However, we did not observe changes in *Dio2* gene expression (*p* = 0.90, Figure 4B). These data suggest that intracellular TH metabolism was unchanged. Also, no differences were observed in *Thra1* and *Hr* mRNA levels during late myogenic differentiation (*p* = 0.30, Figure 4C and *p* = 0.08, Figure 4D).

## 3. Discussion

A proinflammatory wave marks the initial muscle regeneration phase in vivo. At the same time, the muscle progenitor cells are activated, and the myogenic process starts [22]. However, if the proinflammatory microenvironment persists, it can disturb myogenic progression [21,23]. Myoblast/myotube C2C12 cells express Toll-like receptor 4 (TLR4) and respond to LPS by producing proinflammatory mediators such as TNF-α, IL-6, and CCL2 [24,25]. 

Our model used a low LPS dose (10 ng/mL), 10 to 100 times lower than other studies [21]. Although low, the LPS dose significantly increased IL-6, IL-1β, CXCL2, and CCL2 production in C2C12 cells cultured in myoblast conditioned medium, indicating a proinflammatory microenvironment. 

Next, we evaluated the myogenic progression during a persistent proinflammatory environment by analyzing the expression of crucial myogenic regulator genes. Our findings showed that LPS significantly reduced MyoD transcription during initial differentiation (24 h after differentiation), suggesting a reduction in the initial myogenic stimulus in C2C12 myoblasts. Our data corroborate the results obtained by Ono and Sakamoto [21], who observed a significant decrease in MyoD protein levels in differentiating myoblasts C2C12 treated with 100 or 1000 ng/mL LPS [21].

Previous studies have shown that MyoD^−/−^ mice primary myoblasts fail to induce the expression of MyoG, resulting in impaired differentiation and regeneration [26]. However, we did not observe a reduction in *Myog* and MyHC (*Myh7* and *Myh4*) expression in late differentiated myocytes, as expected. Furthermore, it is also essential to notice that in the absence of MyoD in vitro, the myogenin expression can be induced by other MRF such as Myf5 [26]. After the MyoD increase, myogenin is the next key MRF in the myogenic process [26]. Myogenin induces myosin and other genes involved in late differentiation, leading to myofiber maturation [26]. Since *Myog* expression does not alter in early myogenesis in our data, it was expected that the myosin expression (*Myh4* and *Myh7*) would not change in late myogenesis. Together, these data suggest that the impact of a low inflammatory milieu is limited. On the other hand, Ono and Sakamoto observed a reduction in the protein levels of MyoG and MyHC IIb in a dose-dependent manner during late differentiation (100 or 1000 ng/mL LPS at 144 h after induction of differentiation) [21]. Therefore, the difference between both studies could be due to the difference in LPS doses used.

The Myod, Myogenin protein levels, and mRNA are synchronized during myogenesis [26]. Thus, when their gene expression is altered, the modulations are in the same direction, and similar intensity is observed at protein levels [27]. Therefore, our qPCR results regarding Myod and myogenin expression can be compared with the protein levels from other studies. However, this is not always the case, thus, the mRNA levels and proteins activity can be regulated in opposite directions, as in the *Dio2* expression and D2 activity [8].

T3 signaling is necessary for MyoD expression in early myoblast [11,14]. Additionally, systemic inflammatory conditions can reduce skeletal muscle TH response [16]. *Thra1* expression was decreased in the muscle of chronic inflamed and septic mice [16,18]. The reduction of *Thra1* expression, through silencing in C2C12 myoblasts, impaired myoblast proliferation and differentiation, compromising myogenesis [14]. However, we did not observe changes in *Thra1* expression in early or late differentiation.

Nevertheless, T3 action depends on THRs occupancy rate, which depends both on the number of receptors and the T3 intracellular levels. The intracellular levels of T3 are dynamic, as they depend both on its influx/efflux and the intracellular metabolism of TH by deiodinases. Thus, the unchanged *Thra1* expression does not mean that LPS does not alter T3 action.

MCT8 is the primary TH transporter in the skeletal muscle. Herein we evaluated the gene expression of *Slc16a2*, which encodes MCT8. Myocytes express more MCT8 than myoblasts [13]. During the late differentiation phase, LPS increased *Slc16a2* expression compared to control myocytes. However, we did not observe changes in early differentiation. This data suggests that LPS increases the TH flow through the sarcolemma only in late differentiated cells since MCT8 promotes both the entry and exit of HTs from the cell. 

During myogenesis, the expression of D2 and D3 is finely regulated. The inactivation of TH by D3 is necessary for the survival and proliferation of satellite cells [10] while T3 production by D2 is fundamental for muscle differentiation in neonates and C2C12 myoblasts in vitro [11,28]. Herein, we observed that the LPS-induced proinflammatory microenvironment reduces *Dio2* expression in myoblasts. At the same time, LPS increased D2 enzymatic activity during early differentiation, suggesting an increase in intracellular T3. The *Dio2* gene is positively responsive to NFkB pathway activation by LPS [29]. *Dio2* is negatively regulated by T3, whereas D2 activity is reduced by the substrate T4 [30]. The increase in D2 activity suggests an increase in intracellular T3, which, in a way, could explain the reduction in *Dio2* mRNA found in the same stage of myogenesis. The possible increase in T3 level may also explain the resumption of myogenic stimulus since increased D2 activity in myoblasts could accelerate myogenic differentiation. Additionally, the increase in D2 activity may be a compensatory response that kept TH signaling intact and possibly involved in the recovery of the myogenic program.

Our data analyzed the impact of the low inflammatory milieu in myogenesis in an in vitro model. In elderly and obese subjects, it is common to observe a low chronic inflammation [31,32]. In both obesity and aging patients, decreased skeletal muscle regeneration is associated with reduced myogenesis [33,34]. However, the molecular mechanism related to low levels of chronic inflammatory stimulus that could modulate the myogenic program is not entirely understood. In both aged and obese adults, a high frequency of subclinical hypothyroidism is observed in clinical practice [33,35]. Considering the importance of TH to myogenesis [36], it is imperative to understand better how low inflammatory stimulus could change the intracellular T4 and T3 metabolism during myogenesis. Herein, we observed that the decrease in *Myod* in early differentiation period does not significantly affects the late myotube differentiation. However, we did observe a modulation in TH metabolism in early and late differentiated cells, suggesting that inflammation does affect T3 delivery to the nucleus. 

In our in vitro experiments studying the effects of an inflammatory challenge during myogenesis, we explored a model related to the early regeneration process since we used myoblasts under differentiation to myocytes. Our model is related to the end of the first inflammatory wave and initiation of the resolution process, and the second proinflammatory wave was characterized by T cell and M1 macrophage influx [37]. 

However, our study presents some limitations. First of all, we use an in vitro model with the C2C12 cells that is validated to study myogenesis but not to evaluate muscle functional alterations. In addition, during myogenesis in vivo, the muscle progenitor cells interact with systemic factors and other cell types such as mature myofibers, fibroblast, and inflammatory cells [3]. Although, the in vitro myogenesis model is largely used, the interaction with systemic factors and other cells type is lost as in any cell study. Another important limitation is the lack of D3 evaluation in our study due to the difficulty of the technique caused by low expression levels of this gene. The D3 activity analysis could help in the understanding of TH intracellular homeostasis. Thus, D3 data, together with intracellular TH content, could provide better information about TH inactivation and nuclear levels during myogenesis under low-grade inflammatory conditions.

To measure responsiveness to T3 during myogenesis, we evaluated the expression of *Hr*, a T3 target gene. *Hr* was one of the first genes described as responsive to T3 [38]. The skeletal muscle *Hr* expression has already been identified and used as a parameter of T3 responsiveness [13,39]. We did not find a difference in *Hr* expression in myoblast or myocytes/myotubes under LPS stimulus, suggesting that TH signaling was preserved during myogenesis even under the low inflammatory condition.

Thus, herein we demonstrated that low inflammatory activity could not disrupt TH intracellular signaling in myoblasts, which could explain the recovery of the myogenic program in myocytes exposed to LPS during early differentiation.

## 4. Materials and Methods

### 4.1. Cell Culture

Proliferative myoblasts C2C12 (CRL-1772™—ATCC—Manassas, VI, USA; donated by Anita Boelen) in passage number 7 to 10 were expanded in growth medium (Dulbecco’s Modified Eagle Medium, DMEM (Gibco, #12800-017, Thermo Fisher Scientific, Waltham, MA, USA) supplemented with 10% fetal bovine serum (FBS, #12306C, Sigma Aldrich, Saint Luis, MO, USA), 1% penicillin, and 1% streptomycin (PS—#15140, Gibco, Thermo Fisher Scientific, Waltham, MA, USA) at incubator atmosphere 5% CO_2_ and 37 °C. The cells were seeded in 0.1% gelatin 6 well plates (Costar, Corning Glandale, AZ, USA) at 5.23 × 10^3^ cells/cm^2^ and cultured in growth medium for 48 h until reaching 80–100% confluence. On day 0, the differentiation was induced by medium change (DMEM supplemented with 2% horse serum (HS—Sigma Aldrich, Saint Luis, MO, USA), 1% penicillin, 1% streptomycin). The control group only received the medium, while the experimental group was cultured in medium with 10 ng/mL of lipopolysaccharide (LPS—E. coli O127:B8, Sigma Aldrich, Saint Luis, MO, USA). The medium of both groups was refreshed every 48 h. The conditioned medium was collected at 24 h. Cell samples were collected in cold PBS on days 1 (24 h) and 4 (96 h) after differentiation induction by cell scrap.

### 4.2. RNA Extraction and Gene Expression (qPCR)

The cell samples were centrifuged at 500 g for 5 min. The cell pellet was homogenized in TRIzol® Reagent (Thermo Fisher Scientific, Waltham, MA, USA) and stored at −20 °C until RNA extraction following the manufacturer’s protocol. RNA amount and purity RNA (260/280 and 260/230 ratios) were measured by nano spectrophotometer (Inplen, Westlake Village, CA, USA). cDNA was synthesized from a 1000 ng RNA input using the High-Capacity cDNA Reverse Transcription kit following the manufacturer’s instructions (Applied Biosystems, Waltham, MA, USA). For qPCR analysis, EvaGreen^®^ qPCR SuperMix (Solis BioDyne, Tartu, Estonia) and intronspaning primers were used (Appendix A). The qPCR reaction was performed in the Eppendorf Mastercycler^®^ RealPlex. The qPCR reaction efficiency and melting curve were analyzed for each assay. Gene expression was quantified using the standard curve method, and the target gene expression was normalized by the arithmetic mean of reference gene expression: *Hprt1* (HPRT) e *G6pd2* (G6PDH).

### 4.3. Deiodinase Activity

C2C12 cells were removed from the culture flasks using cell scrapers, washed with PBS, and centrifuged at 500 g for 5 min. Cell pellets were stored at −80 °C until the assay was performed at the Thyroid Unit, Hospital de Clínicas de Porto Alegre. D2 activity was measured as previously described [40]. Briefly, the cells were sonicated in 0.25 M sucrose in PE buffer (0.1 M potassium phosphate, 1 mM EDTA), 10 mM dithiothreitol—DTT). The cell lysate (100 μg) was incubated with 100,000 CPM of (^125^I) T4, 20 mM DTT, and 4 nM T4 in the final volume of 300 μL of PE buffer. The solution was incubated for 120 min at 37 °C. Subsequently, the ^125^I released in the deiodination reaction was separated from the labeled T4 by the addition of 200 μL of horse serum and 100 μL of 50% trichloroacetic acid, followed by centrifugation for 2 min at 12,000× g. The generated ^125^I was quantified using the 2427 Wizard2 automatic gamma counter (PerkinElmer, Waltham, MA, USA). D2 activity was expressed in de-iodined T4 femtomolar per minute per mg of protein.

### 4.4. Cytokines and Chemokines Measurement

The C2C12 myoblast conditioned medium samples of the first experiments (n = 5) were collected and stored at −20 °C until cytokine concentration measurements were performed. IL-6, IL-1β, and chemokines CCL-2, and CXCL-1 were quantified using the MILLIPLEX-MAP Mouse Cytokine/Chemokine Magnetic Bead Panel—Immunology Multiplex Assays kit (MCYTOMAG-70K; Merck Millipore, MA, USA) according to the manufacturer’s protocol. The plate was read on the MAGPIX® System equipment (Merck Millipore, Saint Luis, MO, USA), which provided each analyte′s median fluorescence intensity in the sample. The levels of each analyte were calculated using the Luminex xPonent^®^ for MAGPIX^®^ v software 4.2 (Luminex Corporation, Austin, TX, USA) using the standard curve method.

### 4.5. Statistical Analysis

Results are presented as mean ± standard deviation or median and quartiles of the mean of at least five independent experiments. The D’Agostino and Pearson omnibus test was performed to evaluate the parametric or nonparametric data distribution. Parametric data were analyzed by Student’s T-test, and nonparametric data by Mann–Whitney U test. The difference between the groups was considered when the p-value was less than 0.05. Statistical analysis was performed using the GraphPad Prism 6 software (GraphPad Software, Inc., San Diego, CA, USA).

## 5. Conclusions

We demonstrated that a low LPS dose induced myoblasts to produce cytokines and chemokines typical of an inflammatory response. This milieu reduced the initial myogenic stimulus and differently modulated the D2 expression and action of in myoblasts and myocytes. However, the LPS challenge did not delay late differentiation. These data suggest that low-grade chronicle inflammation modulates intracellular myoblast response and influences myogenic homeostasis. Herein, the low-grade inflammatory stimulus could impact skeletal muscle regeneration affecting aged and obese patients.

## Figures and Tables

**Figure 1 metabolites-12-00416-f001:**
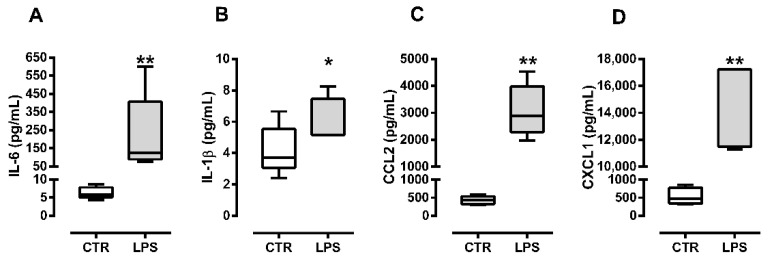
C2C12 myoblasts 24 h post differentiation induction respond to LPS by producing proinflammatory cytokines and chemokines. IL-6 (**A**), IL-1β (**B**) cytokines and CCL2 (**C**), CXCL-1 (**D**) chemokines concentration in myoblast conditioned medium 72 h after first LPS stimulus (10 ng/mL). The white bar represents the control group (CTR), and the gray bar represents the LPS group. Data are expressed as median and quartiles of five independent experiments. The normal distribution was not expected due to the small sample size (*n* = 5); the nonparametric test Mann–Whitney U was applied. * *p* < 0.05, ** *p* < 0.01.

**Figure 2 metabolites-12-00416-f002:**
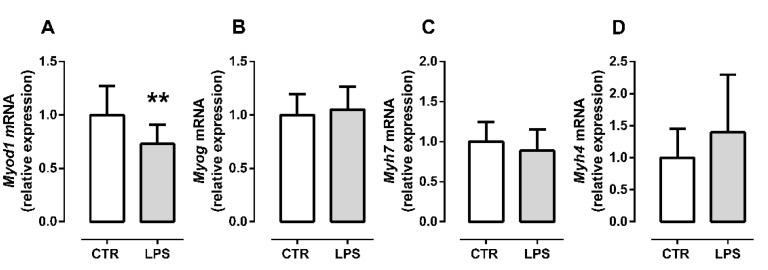
LPS administration affects initial myogenic stimulus but not late differentiation. *Myod1* expression at 24 h post differentiation induction (**A**) and expression of *Myog* (**B**), *Myh7* (**C**), and *Myh4* (**D**) in C2C12 myocytes/myotubes mixed culture at 96 h post differentiation induction. The white bar represents the control group (CTR), and the gray bar represents the LPS group (LPS, 10 ng/mL). Data are expressed as mean ± SD of 12 independent experiments. The data normal distribution was confirmed by D’Agostino and Pearson omnibus test; the parametric Student’s *t*-test was applied. ** *p* < 0.01.

**Figure 3 metabolites-12-00416-f003:**
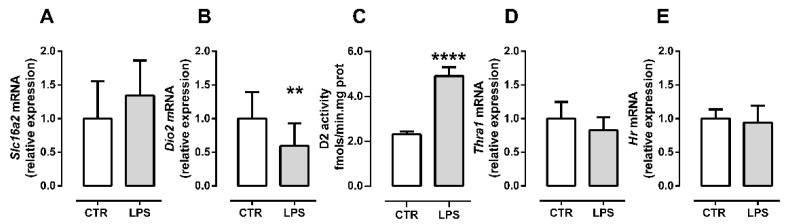
LPS alters TH metabolism in C2C12 myoblasts during initial differentiation. Expression *Slc16a2* (**A**), *Thra1* (**B**), *Dio2* (**C**), D2 activity (**D**), and expression of the TH responsive-gene *Hr* (**E**) in C2C12 myoblast culture at 24 h post differentiation induction. The white bar represents the control group (CTR), and the gray bar represents the LPS group (LPS, 10 ng/mL). Data are expressed as mean ± SD of at least eight independent experiments in (**A**,**B**,**D**,**E**) and five independent experiments in **C**. The data normal distribution was confirmed by D’Agostino and Pearson omnibus test; the parametric Student’s *t*-test was applied. ** *p* < 0.01, **** *p* < 0.0001.

**Figure 4 metabolites-12-00416-f004:**
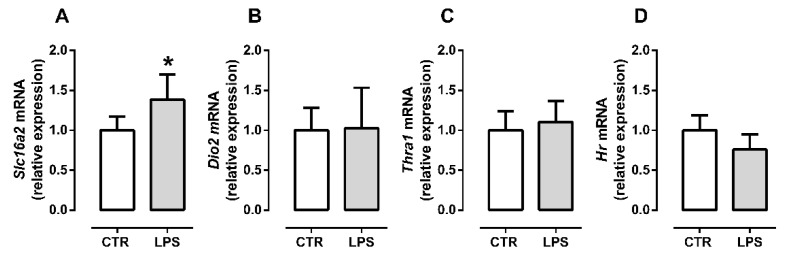
LPS increases TH transporter expression in C2C12 myocytes/myotubes during late differentiation. Expression *Slc16a2* (**A**), *Dio2* (**B**), *Thra1* (**C**), and expression TH responsive-gene *Hr* (**D**) in C2C12 myocytes/myotubes mixed culture at 96 h post differentiation induction. The white bar represents the control group (CTR), and the gray bar represents the LPS group (LPS, 10 ng/mL). Data are expressed as mean ± SD of at least five independent experiments The data normal distribution was confirmed by D’Agostino and Pearson omnibus test; the parametric Student’s *t*-test was applied. * *p* < 0.05.

## Data Availability

The data presented in this study are available within the article and will be provided by the correspondent author under reasonable request.

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
