# Peer review of "Low Inflammatory Stimulus Increases D2 Activity and Modulates Thyroid Hormone Metabolism during Myogenesis In Vitro"

_metabolites, 2022, doi:10.3390/metabo12050416_

Round 1

Reviewer 1 Report

This manuscript by Oliveria, et al. is a descriptive analysis of the impact of LPS stimulation on a few transcripts associated with muscle differentiation in C2C12 cells. Overall, the data are sound, the methods are appropriate, and the authors (mostly) interpret the data in an appropriate manner.

The authors demonstrate that LPS stimulation induces C2C12 cells to produce relevant pro-inflammatory cytokines and chemokines, confirming the C2C12 model as a cell line that initiates a typical physiological response to LPS. The authors then demonstrate a modest reduction of Myod1 transcripts following LPS administration. However, they show that this modest and strictly statistically significant change in Myod1 expression is likely not biologically relevenat as Myog, Myh7, and Myh4 show no (statistically significant) change. The authors further show a (statistically significatn) decrease in Dio2 transcripts that is, interestingly, concomitant with a clear increase in D2 activity, but no associated change in Thra1 or Hr expression. Finally, an increase in TH transporter expression is observed during late differentiation. 

As mentioned, the experiments/methods appear sound. However, this reviewer urges the authors to be more forthright in how they interpret their data. The authors seem to only care about p values associated with Student's t-tests to the exclusion of obvious biological significance. For example, while they label the change in Myod1 expression as significant, they discount any cange in Myh4, Slc16a2, Thra1(early), or Hr (late) expression. The inherent noisiness of the authors' data are clearly driving what they do and do not consider significant, and thus the authors are only considering statistics and not biological relevance. However, this is not an honest evaluation of the data as the "significant" change in Myod1 is clearly not biologically relevant as shown by their lack of change (or even clear increase) for the other transcripts in Figure 2. 

Also, the authors seem to be strictly considering mRNA transcripts rather than protein product. It is totally fine to publish a paper whose data are exclusively transcript levels. However, the results must be interpreted in the context of a lack of information about the protein products. For example, the studies cited by this manuscript assess both mRNA and protein expression, and therefore the authors' findings in this paper should be discussed in the context of possible protein changes not detected by their sole focus on transcripts. 

Therefore, overall, the data of this paper, while incremental and purely confirmatory of previous publications, is fine for this journal. However, this reviewer recommends that the authors modify how they describe their results with respect to relying soley on a t-test to describe whether or not a change has actually occured, and also bringing their lack of protein products into account in their interpretation. After they have made these changes, then this manuscript should be acceptable for publication. 

Author Response

Response to reviewer point by point:

Reviewers’ comment are in bold text. Authors’ answers are in italic text.

Reviewer 1 - Q&A

This manuscript by Oliveria, et al. is a descriptive analysis of the impact of LPS stimulation on a few transcripts associated with muscle differentiation in C2C12 cells. Overall, the data are sound, the methods are appropriate, and the authors (mostly) interpret the data in an appropriate manner.

Answer: We thank the reviewer 1 for the careful reading and suggestions below that helps to improve our manuscript.

Reviewer 1: The authors demonstrate that LPS stimulation induces C2C12 cells to produce relevant pro-inflammatory cytokines and chemokines, confirming the C2C12 model as a cell line that initiates a typical physiological response to LPS. The authors then demonstrate a modest reduction of Myod1 transcripts following LPS administration. However, they show that this modest and strictly statistically significant change in Myod1 expression is likely not biologically relevenat as Myog, Myh7, and Myh4 show no (statistically significant) change. The authors further show a (statistically significatn) decrease in Dio2 transcripts that is, interestingly, concomitant with a clear increase in D2 activity, but no associated change in Thra1 or Hr expression. Finally, an increase in TH transporter expression is observed during late differentiation. 

Answer: We would like to thank reviewer 1 to give us the opportunity to further discuss our data. We agree with reviewer 1 regarding the effects observed in the early myogenesis do not extend to a late stage. The Myog expression is induced by Myod1, however, in the absence of MyoD the myogenin expression can be induced by other MRF as Myf5 (Dumont et al., 2015). Since myogenin is the next MRF in the myogenic process (Dumont et al., 2015) and its expression is maintained in early myogenesis, it was expected that the myosin expression (Myh4 and Myh7) would not be altered. Suggesting that the impact of a low inflammatory milieu is limited. This information is new in the literature and gives insights to understand the impact of low-grade inflammation.  In order to turn our text clearly, we change lines on pages 06 (please, see below). Additionally, the increase in D2 activity suggests an increase in local triiodothyronine, since the Thra expression did not change, it is expected an increase in thyroid hormone receptor occupancy. Although HR did not change, the decrease in Dio2 expression strongly suggests an increase in T3 action. It is important to notice that Dio2 gene is rapidly modulated by fluctuations in intracellular T3 levels. Dio2 is negatively regulated by T3. Thus, an increase in TH signaling could induce myogenin expression by itself (Ambrosio et al., 2017; Sagliocchi et al., 2019).

Lines from page 6: “(...) Furthermore, it is also essential to notice that in the absence of MyoD in vitro, the myogenin expression can be induced by other MRF such as Myf5 [26]. After the MyoD increase, myogenin is the next key MRF in the myogenic process [26]. Myogenin induces myosin and other genes involved in late differentiation, leading to myofiber maturation [26]. Since, in our data, Myog expression didn't alter in early myogenesis, it was expected that the myosin expression (Myh4 and Myh7) would not change in late myogenesis. Together, these data suggest that the impact of a low inflammatory milieu is limited.  (...)”

As mentioned, the experiments/methods appear sound. However, this reviewer urges the authors to be more forthright in how they interpret their data. The authors seem to only care about p values associated with Student's t-tests to the exclusion of obvious biological significance. For example, while they label the change in Myod1 expression as significant, they discount any cange in Myh4, Slc16a2, Thra1(early), or Hr (late) expression. The inherent noisiness of the authors' data are clearly driving what they do and do not consider significant, and thus the authors are only considering statistics and not biological relevance. However, this is not an honest evaluation of the data as the "significant" change in Myod1 is clearly not biologically relevant as shown by their lack of change (or even clear increase) for the other transcripts in Figure 2. 

Answer: We appreciated the reviewer point of view. However, this is a controversial way to describe/discuss anyones' data. If we had decided to discuss the increases and decreases without taking in account the statistical evaluation, we would have been criticized as well.  Therefore, we decided to only discuss the data with statistical significance.The discussion about statistical and biological significance is extremely important and fruitful (Di Leo and Sardanelli, 2020; Martínez-Abraín, 2008; EFSA   Scientific   Committee, 2011). The authors only found relevant to discuss the report with statistical support, to discard the null hypothesis in order to avoid biased  /  unreliable  results. As suggested in the literature, we decided, before the experiment, that biological significance will be discussed if p value were between 0.06 and 0.05. The  Myh4, Slc16a2 (early), Thra1(early), or Hr (late) expression had p<0.06. However, in order to provide the readers a better opportunity to take their own conclusion, we added the p values in the text (please see the revised manuscript). We improved our figure and legend to help the readers to better visualize our data (figure 2). Regarding Myod1 decreased expression do not present a biological role, it is not possible to authors argue in this favor. The low inflammatory milieu can be observed during aging (inflammaging) and in obesity (low chronic inflammation) in both conditions patients can present decrease in skeletal muscle regeneration (Bloise et al., 2018; Sanyal and Raychaudhuri, 2016). Furthermore, low and high degrees of inflammatory stimulus can produce different responses  (Bloise et al., 2016). Therefore, a better understanding of the molecular pathways associated with low levels of inflammatory stimulus on myogenis is important and could help in the treatment development in both conditions. We also included this discussion at page 7, please see below.

Lines from page 7: “Our data analyzed the impact of the low inflammatory milieu in myogenesis in an in vitro model. In elderly and obese subjects, it is common to observe a low chronic inflammation [31,32].  Both obesity and aging patients decrease skeletal muscle regeneration is associated with reduced myogenesis [33,34]. However, the molecular mechanism related to low levels of chronic inflammatory stimulus that could modulate the myogenic program is not entirely understood. In both aged and obese adults, a high frequency of subclinical hypothyroidism is observed in clinical practice [33,35]. Considering the importance of TH to myogenesis [36], it is imperative to understand better how low inflammatory stimulus could change the intracellular T4 and T3 metabolism during myogenesis. Herein, we observed that the decrease in Myod in early differentiation period does not significantly affects the late myotube differentiation. However, we did observe a modulation in TH metabolism in early and late differentiated cells, suggesting that inflammation does affect T3 delivery to the nucleus.”

Also, the authors seem to be strictly considering mRNA transcripts rather than protein product. It is totally fine to publish a paper whose data are exclusively transcript levels. However, the results must be interpreted in the context of a lack of information about the protein products. For example, the studies cited by this manuscript assess both mRNA and protein expression, and therefore the authors' findings in this paper should be discussed in the context of possible protein changes not detected by their sole focus on transcripts. 

Answer: We appreciate reviewer 1 comment. Protein level analysis is extremely important, especially when protein activity and expression are desynchronized, as in the case of D2 activity and Dio2 mRNA expression. The Myod, Myogenin mRNA are modulated in the same direction and similar intensity to the mRNA and protein levels during myogenesis (Ferri et al., 2009). In this sense, we did not understand that investigating Myod or myogenin expression would be crucial for understanding biological actions. Additionally, myosin expression in the manuscript context is important as a marker of myoblast differentiation into myotubes, the protein contract action itself does not provide valuable biological information. Thus, analyzing the myosin protein levels, seems not relevant for this study. In order to increase the comprehension of this subject, we include the discussion that the protein levels could be modulated as in previous work at the fourth paragraph from page 6.

Fifth paragraph from page 6: “The Myod, Myogenin protein levels, and mRNA are synchronized during myogenesis [26]. Thus, when their gene expression is altered, modulations in the same direction, and similar intensity is observed at protein levels [27]. Therefore, our qPCR results regarding Myod and myogenin expression can be compared with the protein levels from other studies. However, this is not always the case, thus, the mRNA levels and protein activity can be regulated in opposite directions, as in the Dio2 expression and D2 activity [8].”

Therefore, overall, the data of this paper, while incremental and purely confirmatory of previous publications, is fine for this journal. However, this reviewer recommends that the authors modify how they describe their results with respect to relying soley on a t-test to describe whether or not a change has actually occured, and also bringing their lack of protein products into account in their interpretation. After they have made these changes, then this manuscript should be acceptable for publication. 

Answer:  We thank reviewer 1 to help us to improve our manuscript. We included the discussion suggested in the lines mentioned above.

References:

Ambrosio R, De Stefano MA, Di Girolamo D & Salvatore D 2017 Thyroid hormone signaling and deiodinase actions in muscle stem/progenitor cells. Molecular and Cellular Endocrinology 459 79–83. (doi:10.1016/j.mce.2017.06.014)

Bloise FF, Van Der Spek AH, Surovtseva O V., Ortiga-Carvalho TM, Fliers E & Boelen A 2016 Differential Effects of Sepsis and Chronic Inflammation on Diaphragm Muscle Fiber Type, Thyroid Hormone Metabolism, and Mitochondrial Function. Thyroid 26 600–609. (doi:10.1089/thy.2015.0536)

Bloise FF, Oliveira TS, Cordeiro A & Ortiga-Carvalho TM 2018 Thyroid hormones play role in sarcopenia and myopathies. Frontiers in Physiology 9 1–7. (doi:10.3389/fphys.2018.00560)

EFSA Scientific Committee 2011 Statistical Significance and Biological Relevance. EFSA Journal 9. (doi:10.2903/j.efsa.2011.2372)

Ferri P, Barbieri E, Burattini S, Guescini M, D’Emilio A, Biagiotti L, Del Grande P, De Luca A, Stocchi V & Falcieri E 2009 Expression and subcellular localization of myogenic regulatory factors during the differentiation of skeletal muscle C2C12 myoblasts. Journal of Cellular Biochemistry 108 1302–1317. (doi:10.1002/jcb.22360)

Franceschi C, Garagnani P, Parini P, Giuliani C & Santoro A 2018 Inflammaging: a new immune–metabolic viewpoint for age-related diseases. Nature Reviews Endocrinology 14 576–590. (doi:10.1038/s41574-018-0059-4)

Kalinkovich A & Livshits G 2017 Sarcopenic obesity or obese sarcopenia: A cross talk between age-associated adipose tissue and skeletal muscle inflammation as a main mechanism of the pathogenesis. Ageing Research Reviews 35 200–221. (doi:10.1016/j.arr.2016.09.008)

Kawai T, Autieri M V. & Scalia R 2021 Adipose tissue inflammation and metabolic dysfunction in obesity. American Journal of Physiology-Cell Physiology 320 C375–C391. (doi:10.1152/ajpcell.00379.2020)

Di Leo G & Sardanelli F 2020 Statistical significance: p value, 0.05 threshold, and applications to radiomics—reasons for a conservative approach. European Radiology Experimental 4 18. (doi:10.1186/s41747-020-0145-y)

Martínez-Abraín A 2008 Statistical significance and biological relevance: A call for a more cautious interpretation of results in ecology. Acta Oecologica 34 9–11. (doi:10.1016/j.actao.2008.02.004)

Sagliocchi, S.; Cicatiello, A.G.; Di Cicco, E.; Ambrosio, R.; Miro, C.; Di Girolamo, D.; Nappi, A.; Mancino, G.; De Stefano, M.A.; Luongo, C.; et al. 2019 The thyroid hormone activating enzyme, type 2 deiodinase, induces myogenic differentiation by regulating mitochondrial metabolism and reducing oxidative stress. Redox Biol., 24, (doi:10.1016/j.redox.2019.101228.)

Sanyal D & Raychaudhuri M 2016 Hypothyroidism and obesity: An intriguing link. Indian Journal of Endocrinology and Metabolism 20 554. (doi:10.4103/2230-8210.183454)

Reviewer 2 Report

In the manuscript "LPS-induced inflammatory milieu affect thyroid hormone metabolism and impaired myogenesis in time dependent manner", Thamires Siqueira de Oliveira and colleagues performed study on the LPS-induced inflammatory milieu on the thyroid hormone metabolism. However, some minor criticisms are present, as follows:

- the chapters are not in order, after results we have discussion and after material and methods and the conclusions. Please correct that

- The title should be shorter and clinical based

- Clinical implications of the study should be emphasized in the thesis and conclusions  

Author Response

Reviewer 2- Q&A

Reviewers’ comment are in bold text. Authors’ answers are in italic text.

In the manuscript "LPS-induced inflammatory milieu affect thyroid hormone metabolism and impaired myogenesis in time dependent manner", Thamires Siqueira de Oliveira and colleagues performed study on the LPS-induced inflammatory milieu on the thyroid hormone metabolism. However, some minor criticisms are present, as follows:

Answer: The authors thank reviewer 2 for the time spend to analyze our manuscript and for the insightful suggestions.

- the chapters are not in order, after results we have discussion and after material and methods and the conclusions. Please correct that

Answer: The authors used the Metabolites’ template provided in the journal website. The order from the template is:  1. Introduction; 2. Results; 3. Discussion; 4. Materials and Methods; 5. Conclusions.

- The title should be shorter and clinical based

Answer: We thank reviewer 2 for this suggestion. We changed our title from: “LPS-induced inflammatory milieu affect thyroid hormone metabolism and impaired myogenesis in time dependent manner” to “Low inflammatory stimulus affects thyroid hormone metabolism during myogenesis in vitro ”

- Clinical implications of the study should be emphasized in the thesis and conclusions

Answer: We thank reviewer 2 for this comment. Indeed, our manuscript should include more of the clinical implications. In this sense, we added the discussion at page 7 and page 9 in the discussion and conclusion, respectively.

Lines from page 7: “Our data analyzed the impact of the low inflammatory milieu in myogenesis in an in vitro model. In elderly and obese subjects, it is common to observe a low chronic inflammation [31,32].  Both obesity and aging patients decrease skeletal muscle regeneration is associated with reduced myogenesis [33,34]. However, the molecular mechanism related to low levels of chronic inflammatory stimulus that could modulate the myogenic program is not entirely understood. In both aged and obese adults, a high frequency of subclinical hypothyroidism is observed in clinical practice [33,35]. Considering the importance of TH to myogenesis [36], it is imperative to understand better how low inflammatory stimulus could change the intracellular T4 and T3 metabolism during myogenesis. Herein, we observed that the decrease in Myod in early differentiation period does not significantly affects the late myotube differentiation. However, we did observe a modulation in TH metabolism in early and late differentiated cells, suggesting that inflammation does affect T3 delivery to the nucleus.”

Lines from page 9: “(...) These data suggest that low-grade chronicle inflammation modulates intracellular myoblast response and influences myogenic homeostasis. Thus, the low-grade inflammatory stimulus could impact skeletal muscle regeneration affecting aged and obese patients. (...)”

Reviewer 3 Report

Paper titled (LPS-induced inflammatory milieu affect thyroid hormone metabolism and impaired myogenesis in time dependent manner) by  de Oliveira et al.

1- Abstract: s is unclear should be "needs further investigation".

2- Revise figure legends for figure 2, mainly p value & sympols need to be larger and clearer for all figures

3- Why some data analyzed by Mann Whitney U test & others by student t test? explain in figure legends.

4- Submitted  to .... test: does not make sense

5- Give the origin of all chemicals completely and consistently (code, company, town, state and country).

6- Figure resolutions need enhancement.

7- Title: should be in "a time dependent manner"

8- How authors ensured the time dependent manner ? this is not clear enough !!

Author Response

Reviewer 3- Q&A

Reviewers’ comment are in bold text. Authors’ answers are in italic text.

Paper titled (LPS-induced inflammatory milieu affect thyroid hormone metabolism and impaired myogenesis in time dependent manner) by  de Oliveira et al.

Answer: The authors appreciated the reviewer 3 comments and suggestions. Thank your help in improving the manuscript.

1- Abstract: s is unclear should be "needs further investigation".

Answer: We thank reviewer 3 for the opportunity to improve our manuscript and for this suggestion. We changed the abstract.

2- Revise figure legends for figure 2, mainly p value & sympols need to be larger and clearer for all figures

Answer: The figures 2, 3 and 4 were modified as instructed, thank you for the comment.

3- Why some data analyzed by Mann Whitney U test & others by student t test? explain in figure legends.

Answer: The explanation of each statistic test were added to figures’ legends. The U test were performed in non-parametric data and T test on parametric data.

4- Submitted  to .... test: does not make sense

Answer: Reviewer 3 is correct, the item 4.5. “Statistical analysis” were rewritten.

5- Give the origin of all chemicals completely and consistently (code, company, town, state and country).

Answer: The information was added.

6- Figure resolutions need enhancement.

Answer: We appreciated the comment and upload better resolution figure images. We believe that the resolution could be lost in the Word file. We also added the high resolution figures at the journal platform.

7- Title: should be in "a time dependent manner"

Answer: Thank you for the suggestion. We changed our title from: “LPS-induced inflammatory milieu affect thyroid hormone metabolism and impaired myogenesis in time dependent manner” to “Low inflammatory stimulus affects thyroid hormone metabolism during myogenesis in vitro”

8- How authors ensured the time dependent manner ? this is not clear enough !!

Answer:  As acknowledged by the reviewer, the time dependency is not well established. Thus, we tone down this term and focus the discussion regarding the effects during initial and late myogenesis.

Round 2

Reviewer 3 Report

The revised form of paper titled (

1- Figure 1: The data normal distribution could not be tested; this does not make sense. Better say "not expected".

2- Figure 1: since data are not in normal dist: should be demonstrated as median and quartiles (not mean&SE)

3- All other figures & methods: present data as mean &SD (not SE)

4- Title: "affects" need to be replaced by a more meaningful word (increase or decrease).

5- Improve figure resolution.

Author Response

Response to reviewer point by point:

Reviewers' comments are in bold text. The authors' answers are in italic text.

We thank the reviewer for the comments and time to analyze our manuscript.

1- Figure 1: The data normal distribution could not be tested; this does not make sense. Better say "not expected".

Answer: We changed the legend as suggested by the reviewer.

2- Figure 1: since data are not in normal dist: should be demonstrated as median and quartiles (not mean&SE)

Answer: We thank the reviewer for the comment. We changed figure 1 as suggested.

3- All other figures & methods: present data as mean &SD (not SE)

Answer: We update the figures 2, 3, and 4, the respective legends, and the method to use mean & SD.

4- Title: "affects" need to be replaced by a more meaningful word (increase or decrease).

Answer: We changed the title to: "Low inflammatory stimulus increases D2 activity and modulates thyroid hormone metabolism during myogenesis in vitro"

5- Improve figure resolution.

Answer: As previously suggested, we uploaded better-resolution figure images. We used images exported as tiff, with a 1200 dpi resolution. The process of introducing the figures in the Word text may cause the resolution decreases. We also added the journal platform's high-resolution figures in both tiff and pdf formats now.

Round 3

Reviewer 3 Report

Data in figure 2,3,4 are not mean & SD

they are medians and quartiles, please give rational why you draw them like that  or correct the figure to columns representing the mean if they are in normal dist

Correct the figure legends acc to the figures (take the decision)

Author Response

We would like to thank the time and consideration from the reviewer that helped to improve our manuscript. The actualized figures 2, 3, and 4 during R2 were expressed as mean and SD. Previously, in the original manuscript, it was mean and SE. After R2, the only graphics as medians and quartiles are from figure 1, as suggested by the reviewer. Furthermore, all figures' legends are in accordance with the graphic. In the attached file, the reviewer can find the mean and SD data using both bars for figures 2, 3, and 4. Additionally, all the mean and SD of each graphic from figures 1, 2, 3, and 4 are also in a table in the attached file.
